# OptiRSDE: A Novel Approach with Temporal Smoothing and Optimized Feature Matching for Fast and Robust Depth Estimation

## Abstract

Depth estimation accuracy over long ranges is a core problem in robotics, maritime autonomy, terrestrial autonomy, and environmental monitoring, where accurate scene understanding is crucial for safe and informed decision-making. Existing monocular solutions suffer a sharp accuracy drop beyond mid-range, with errors of 10–25% at 50–100 m. Recent deep learning–based stereo networks (e.g., FoundationStereo, DSMNet, MonSter, RAFT-Stereo, CREStereo, Selective-Stereo) achieve impressive results on benchmarks but struggle in real-world extended-range scenarios—frequently collapsing at 20–30 m and beyond, where predictions deviate by factors of 2–3× and object-level depth is often lost. In contrast, a calibrated high-quality stereo system can deliver accurate long-range estimates but at the expense of high computational overhead.

We introduce OptiRSDE (Optimized Robust Stereo Depth Estimation), a lightweight yet robust classical computer vision pipeline that integrates disparity refinement, temporal smoothing, and QR-code–based synchronization. OptiRSDE achieves <3% error at 50 m and 5–10% at 100 m, substantially outperforming both monocular methods and modern deep learning stereo baselines in real-world conditions. Operating at 5 FPS, while requiring only standard chessboard calibration and YOLO-based object detection for deployment. Temporal smoothing and outlier rejection mitigate depth jitter, producing stable long-range depth at object level. Validated on DrivingStereo Yang et al. (2019) and a custom 1080p stereo dataset, our system demonstrates scalable, real-time, extended-range stereo depth estimation—delivering strong generalization where both monocular and state-of-the-art deep learning methods fail.

## 1 Introduction

Real-time, accurate depth estimation is vital for applications like autonomous navigation, robotics, and augmented reality. However, existing vision-based methods exhibit a trade-off between performance and accuracy. Conventional stereo vision approaches, though theoretically precise, are often too slow for real-time use, operating at just 0.2–0.5 fps. To boost frame rates, they often sacrifice accuracy, while modern monocular depth estimation models based on deep learning are too computationally heavy for embedded or mobile systems.

Beyond computational cost, maintaining accuracy and stability in dynamic environments remains a major challenge. Depth precision degrades at long ranges, with typical error rates of 5–10% at 50 meters. Environmental factors worsen this; even minor temperature changes can cause drift and error accumulation of up to 25m at 100m range, requiring frequent recalibration. Temporal instability is another issue, with unsmoothed pipelines showing depth inconsistencies up to ±15% frame-to-frame. This is worsened by stereo camera desynchronization, where a misalignment of 10–12 frames can introduce an additional 10% error at 10–50m distances.

While sensors like LiDAR offer high-accuracy depth data Wang et al. (2020), they come with limitations. High-performance units are too expensive for broad use, while affordable ones have limited range, often below 60 meters. LiDAR also suffers from weather sensitivity and surface reflectiv-

ity issues. Critically, an end-to-end stereo video solution that delivers object-specific depth with dynamic detection and tracking remains largely missing.

To address these challenges, we present a novel, efficient pipeline (Figure 1) that balances accuracy, temporal stability, and real-time performance. Our method synchronizes stereo video streams using QR codes (Figure 5), followed by camera calibration, undistortion, and rectification (Figure 3). We use YOLOv11 Khanam & Hussain (2024) for object detection and BRISK-based feature matching Leutenegger et al. (2011) for robust disparity estimation. Multiple optimizations are introduced to enhance performance. Finally, depth is computed via triangulation (Figure 2), forming a practical, end-to-end solution for real-world deployment.

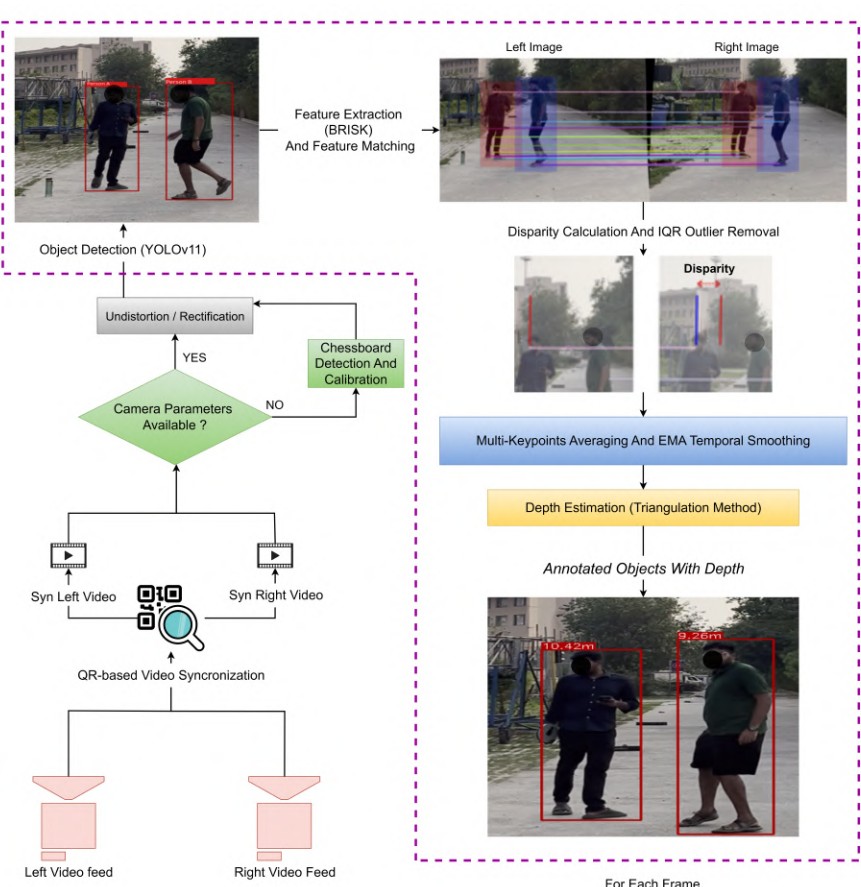

Figure 1: OptiRSDE stereo depth pipeline. Synchronized stereo videos undergo calibration, undistortion, and rectification before object detection and BRISK-based feature matching. Disparity is computed with outlier removal, followed by keypoint averaging and smoothing for depth estimation via triangulation.

## 2 RELATED WORK

Recent advances in depth estimation target computational efficiency, long-range accuracy, and temporal stability; yet, challenges persist for robust, object-specific, real-time depth in constrained environments.

### 2.1 COMPUTATIONAL OVERLOAD AND REAL-TIME PROCESSING

To address the high computational demands in stereo depth rooted in classical correspondence frameworks Scharstein et al. (2001), methods like BiDAStereo Jing et al. (2024) and StereoDiffusion Xu et al. (2024) offer solutions. These rely on robust feature extraction and matching techniques; for

instance, SIFT Lowe (1999) is foundational for establishing correspondences. BiDAStereo employs bidirectional alignment and lightweight recurrent modules, while StereoDiffusion integrates optical-flow-warped disparities into a diffusion model for high-speed, consistent predictions. However, their utility in low-power or embedded systems remains limited and needs further optimization.

## 2.2 LONG-RANGE DEPTH ACCURACY

Long-range depth estimation progresses with trinocular and attention models. For example, Yoshi-hara et al. (2025) localizes vessels up to 2.5 km; GatedStereo Walz et al. (2023) fuses active gated and stereo cues for enhanced performance. While accurate, these systems remain expensive, difficult to deploy, or unsuitable for real-time use. Monocular models like Depth Anything Yang et al. (2024) progress, yet stereo methods like STTR Li et al. (2021a) remain superior for long-range tasks. Large datasets like DrivingStereo Yang et al. (2019) and KITTI Geiger et al. (2012) advance robust mid-to-long range model training. Despite potential, dynamic environment robustness remains challenging.

## 2.3 CALIBRATION AND SYNCHRONIZATION

Accurate stereo depth estimation depends on effective calibration and synchronization. Solutions like BiDAStereo and Li et al. (2021a) use cross-frame alignment, disparity refinement to mitigate misalignment. Optical flow models like CODD Li et al. (2021b) compensate for dynamic motion and drift. Frequent recalibration under changing conditions remains unresolved practically.

## 2.4 TEMPORAL STABILITY

Temporal inconsistencies degrade real-time depth quality. Robust outlier rejection, often via RANSAC Fischler & Bolles (1987), mitigates inconsistencies and noise. Building on linear fil-tering theories Basar (2001), techniques like dual-space disparity refinement Zeng et al. (2025) and DynamicStereo Karaev et al. (2023) significantly reduce variance. StereoDiffusion Xu et al. (2024) incorporates temporal priors for smoothness. While these models reduce jitter, stability in dynamic or low-texture scenes needs improvement.

## 2.5 OBJECT-LEVEL DEPTH ESTIMATION

Object-specific depth estimation gains traction beyond global maps. Systems by Yoshihara et al. (2025) and Zheng et al. (2022) integrate object detection with disparity for tailored depth, e.g., maritime vessels. Transformer pipelines like DynamicStereo Karaev et al. (2023) support object-level consistency by fusing spatial-temporal features. These works emphasize coupling semantic understanding with depth in complex, cluttered scenes.

## 3 METHODOLOGY

The proposed depth estimation pipeline is designed to overcome critical challenges in stereo vision systems, including calibration inaccuracies, unreliable feature detection, instability across frames, and high computational demands. Our approach builds upon and refines the foundational principles of stereo depth estimation.

**Classic Depth Estimation Pipeline Overview:**

- **Calibration**: It establishes intrinsic and extrinsic camera parameters. Intrinsic parameters correct lens distortions (radial, tangential). Extrinsic parameters determine relative camera pose (rotation, translation).

- **Rectification**: Stereo image pairs are transformed for horizontal epipolar line alignment. This rectification Fusiello et al. (2000) simplifies correspondence search as matching pixels lie on the same row.

- **Correspondence Matching**: Feature points are identified and matched across stereo pairs. These matched points generate a disparity map Fua (1991), encoding relative displacement.

### 3.1 STEREO VIDEO SYNCHRONISATION

Temporal alignment of left and right video streams is critical for accurate stereo depth estimation. As our system lacked hardware synchronization capabilities, we implemented a robust software-based method to ensure precise frame-to-frame correspondence. This approach leverages a series of unique visual markers embedded within the video feed to determine and correct any temporal offset. A full description of this synchronization technique is provided in Appendix A.

### 3.2 STEREO CAMERA CALIBRATION

Accurate 3D reconstruction and depth estimation depend on precise stereo camera calibration to determine intrinsic and extrinsic parameters. This process is performed once, after the physical setup is fixed, and must be repeated only if the camera positions change.

Stereo calibration is performed using a chessboard pattern held in front of both cameras. The calibrated parameters are then used for all future video inputs.

This process is shown in detail in Figure 3.

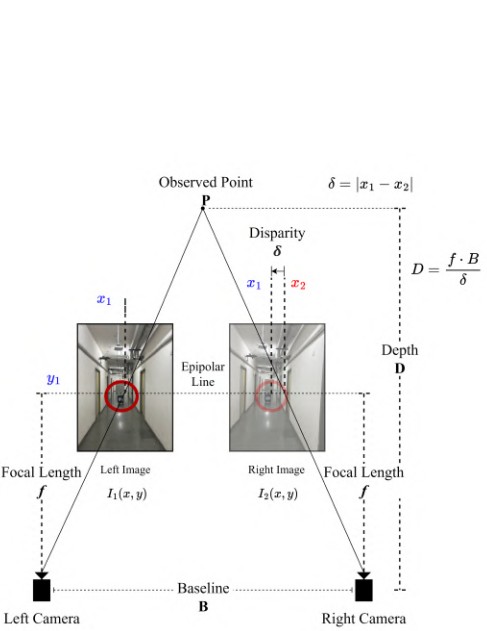

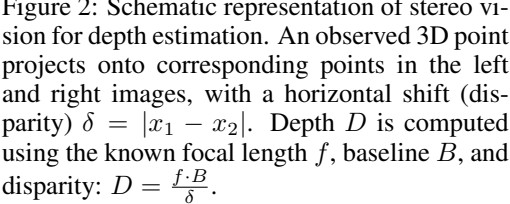

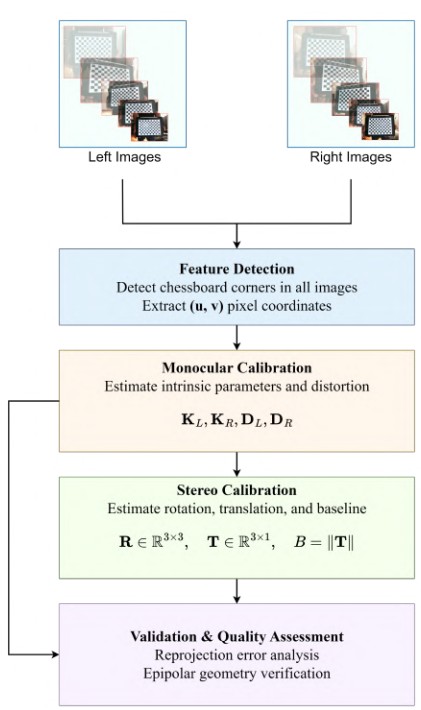

Figure 2: Schematic representation of stereo vision for depth estimation. An observed 3D point projects onto corresponding points in the left and right images, with a horizontal shift (disparity) $\delta = |x_1 - x_2|$. Depth $D$ is computed using the known focal length $f$, baseline $B$, and disparity: $D = \frac{f \cdot B}{\delta}$.

Figure 3: Overview of the stereo camera calibration pipeline. Chessboard corners are detected in synchronized left and right images to extract pixel coordinates. Monocular and stereo calibration estimate intrinsic and extrinsic parameters, followed by validation through reprojection error and epipolar geometry checks.

### 3.3 DEPTH ESTIMATION

Our system estimates depth using disparity-to-depth triangulation (Figure 2) for each detected object. BRISK Leutenegger et al. (2011) is employed for keypoint detection, offering enhanced resilience in low-texture environments compared to traditional methods like ORB Rublee et al. (2011).

This keypoint-based approach is crucial for achieving high accuracy in real-time stereo depth estimation. By restricting keypoint detection to object-specific Regions of Interest (ROIs), the system efficiently prioritizes relevant features for disparity computation. This optimization not only significantly reduces computational load by avoiding unnecessary processing across the entire image but also substantially improves the precision of correspondence matching by adaptively expanding the search space for stereo parallax solely within relevant bounding boxes, thereby enhancing overall system robustness and speed.

Erroneous matches are filtered using the Inter-Quartile Range (IQR). Valid disparities from multiple matched keypoints within each object's bounding box are then averaged for robust depth estimation, similar to harmonic depth averaging Yoshihara et al. (2025), but with a slight numerical precision advantage.

To mitigate frame-to-frame depth fluctuations, Exponential Moving Average (EMA) temporal smoothing is applied every 10 frames. This stabilizes predictions and suppresses noise from spurious and transient errors.

### 3.4 EFFICIENCY-ORIENTED PIPELINE OPTIMIZATIONS

Several optimizations are integrated into the pipeline to balance computational efficiency and estimation accuracy:

- **Keypoint detection is localized** to object bounding boxes through masking, dramatically reducing processing time by avoiding exhaustive search.

- **Feature matching is further constrained** to filtered keypoints of corresponding image regions, yielding $16\times$ speed improvement for this particular component.

- **The FSRCNN-based super-resolution module** Dong et al. (2016), previously used to enhance image detail in Zheng et al. (2022), was eliminated. The use of BRISK compensates for this removal, maintaining robust performance while reducing computational overhead.

### 3.5 TEMPORAL STABILITY IN DYNAMIC ENVIRONMENTS

To ensure temporal consistency in depth estimation within dynamic scenes, the system integrates two complementary techniques:

- Multi-keypoint disparity averaging within object regions, reducing sensitivity to individual erroneous matches.

- EMA-based temporal smoothing, attenuating sudden depth variations from detection noise or transient mismatches. The Exponential Moving Average (EMA) Yu et al. (2020) $\bar{D}_t$ is calculated using the current disparity $P_t$ and the previous smoothed disparity $\bar{D}_{t-1}$:

$$\bar{D}_t = \alpha \cdot P_t + (1 - \alpha) \cdot \bar{D}_{t-1}$$

This strategy, detailed in Algorithm 1, significantly improves the stability and reliability of depth predictions, even under challenging environmental conditions.

## 4 EVALUATION

To evaluate our stereo depth estimation system, we benchmarked OptiRSDE against monocular **(MiDaS)** Ranftl et al. (2022) and stereo (our implementation of Zheng's method Zheng et al. (2022), **BSV Ship**) Zheng et al. (2022) baselines for depth accuracy. Performance comparison is done only between our method and Zheng's method, since it is the only other object-level binocular stereo depth estimation pipeline to the best of our knowledge. We evaluated OptiRSDE on our self-collected dataset (ground truth from distance markers) and DrivingStereo's test images Yang et al. (2024), since it provides LiDAR-based depth maps as ground truth. For our dataset, a reference grid with known ground-truth distances (at 10-meter intervals, using a precise device) was used. Tests were conducted on subjects across **5–100m distances**.

---

**Algorithm 1** Temporal Smoothing for Disparity Estimation

---

**Input**: Left and right frames $F_L, F_R \in \mathbb{R}^{H \times W \times 3}$, current frame number $t \in \mathbb{N}$, smoothing factor $\alpha \in [0, 1]$, expiration threshold $T_{\text{expire}} \in \mathbb{N}$, persistent object state map $S$.
**Output**: Detected objects $B$ (list of bounding boxes), updated disparities $D_{\text{raw}}$.

1:  $B \leftarrow \text{TrackObjects}(F_L)$
2:  $D_{\text{raw}} \leftarrow \text{CalculateDisparities}(F_L, F_R, B)$
3:  $M_{\text{ID} \rightarrow \text{idx}} \leftarrow \emptyset$
4:  **for** each box $b_i \in B$ at index $i$ **do**
5:      $id_{\text{obj}} \leftarrow \text{GetID}(b_i)$
6:      $M_{\text{ID} \rightarrow \text{idx}}[id_{\text{obj}}] \leftarrow i$
7:      **if** $id_{\text{obj}}$ not in $S$ **then**
8:          $S[id_{\text{obj}}] \leftarrow \{\text{disparity: null}, \text{frame: } t\}$
9:      **end if**
10: **end for**
11: **for** each $id_{\text{obj}}$ in keys of $S$ **do**
12:     $s \leftarrow S[id_{\text{obj}}]$
13:     **if** $s.\text{frame} + T_{\text{expire}} \leq t$ **then**
14:         Remove $id_{\text{obj}}$ from $S$
15:         **continue**
16:     **end if**
17:     **if** $id_{\text{obj}}$ in $M_{\text{ID} \rightarrow \text{idx}}$ **then**
18:         $i \leftarrow M_{\text{ID} \rightarrow \text{idx}}[id_{\text{obj}}]$
19:         $d_{\text{current}} \leftarrow D_{\text{raw}}[i]$
20:         **if** $s.\text{disparity} = \text{null}$ **then**
21:             $s.\text{disparity} \leftarrow d_{\text{current}}$
22:         **else if** $d_{\text{current}} = \text{null}$ **then**
23:             $D_{\text{raw}}[i] \leftarrow s.\text{disparity}$
24:         **else**
25:             $d_{\text{smooth}} \leftarrow \alpha \cdot d_{\text{current}} + (1 - \alpha) \cdot s.\text{disparity}$
26:             $s.\text{disparity} \leftarrow d_{\text{smooth}}$
27:             $D_{\text{raw}}[i] \leftarrow d_{\text{smooth}}$
28:         **end if**
29:         $s.\text{frame} \leftarrow t$
30:     **end if**
31: **end for**
32: **return** $B, D_{\text{raw}}$

---

### 4.1 Experimental Settings

### 4.2 Hardware:

The system runs on an AMD Ryzen 5 4600H CPU with 8GB RAM and an NVIDIA GeForce GTX 1650 Ti GPU (4GB VRAM). The stereo vision setup consists of dual 1080p cameras mounted with a fixed 35 cm baseline.

### 4.3 Software:

The system uses Python 3.12 on Linux Mint 22.1 and OpenCV 4.5 for image processing. PyTorch 2.5.1 integrates YOLOv11 for object detection, leveraging CUDA 12.6.

### 4.4 Camera Baseline

Our system uses a 35 cm baseline, balancing flexibility and accuracy, with potential for further extension.

## 4.5 CAMERA CALIBRATION

We used a **9×6 chessboard** across **2 to 20 meters**. Rectification error between stereo pairs was reduced to under 0.75 pixels, ensuring reliable disparity estimation.

## 4.6 OBJECT DETECTION AND STEREO MATCHING

Up to 512 feature points were extracted per frame; only those with $\leq 50$ Hamming distance Liu et al. (2018) were retained.

## 4.7 TEMPORAL SMOOTHING

To ensure consistency, an **Exponential Moving Average (EMA)** ($\alpha = 0.3$) was applied every 10 frames. Outlier disparities were removed using **Inter-Quartile Range (IQR)** Takiar (2023) filtering, reducing noise and stabilizing predictions.

## 5 RESULTS

We compared our method **(OptiRSDE)** against monocular **(MiDaS)** Ranftl et al. (2022) and stereo **(BSV Ship)** Zheng et al. (2022) baselines across two metrics: depth accuracy and speed. Note that we used our own implementation of Zheng's BSV Ship Depth Estimation methodology, since no code is available for it publicly. Tests were conducted on 1080p stereo video at **5–100m distances**. All the results are generated from a single run of the algorithm over each video.

| Method | Err@50m | Err@100m |
|---|---|---|
| Monocular (MiDaS) | 12% (4%) | 25% (9%) |
| Stereo (BSV Ship) | 4% (1.4%) | 10% (6.1%) |
| **OptiRSDE** | **2.8% (0.5%)** | **7.5% (1.9%)** |

Table 1: Depth estimation accuracy comparison. Mean error values with standard deviation in parentheses.

| Method | FPS |
|---|---|
| BSV Ship | 0.3 (0.02) |
| **OptiRSDE** | **5.38 (0.08)** |

Table 2: Execution performance comparison. Mean FPS values with standard deviation in parentheses.

## 5.1 ANALYSIS

We evaluated OptiRSDE's depth estimation accuracy across multiple distances. Estimated distances were validated against calibrated ground truth using our internal dataset and DrivingStereo test images, where LiDAR-based depth maps served as ground truth. Visual results (Figure 4) and quantitative data (Tables 3, 4) are presented. OptiRSDE was compared to our implementation of Zheng's method (*BSV Ship*), MonSter Cheng & et al. (2025), DSMNet Zhang et al. (2020), FoundationStereo Wen et al. (2025), RAFT-Stereo Lipson et al. (2021), CREStereo Li et al. (2022), and Selective-IGEV Wang et al. (2024). Results consistently demonstrate OptiRSDE's superior performance, achieving significantly lower estimation errors, especially at larger distances.

## 5.2 KEY FINDINGS

- **Accuracy:** OptiRSDE reduces errors by 77% vs. MiDaS and 30% vs. BSV Ship at 50m and by 70% vs. MiDaS and 25% vs. BSV Ship at 100m. (Table 1)

- **Speed:** Achieves $> 5$ FPS—10-20× faster than traditional stereo methods. (Table 2)

- **Robustness:** Consistent keypoint detection ensures continuous depth estimation, overcoming frequent detection failures observed in BSV Ship. (See NKD in Table 3)

- **Stability:** Temporal smoothing cuts depth jitter compared to unsmoothed baselines. (Refer to supplementary videos)

| Method | 10m | 20m | 30m | 40m | 50m | 60m |
|---|---|---|---|---|---|---|
| MonSter | -0.68 | -0.6 | +14.22 | +25.4 | -2.2 | +10.76 |
| DSMNet | -0.8 | -1.68 | +7.21 | +17.65 | +15.17 | +30.69 |
| FoundationStereo | -0.77 | -0.81 | **+0.04** | +4.79 | +16.81 | +29.89 |
| RAFT-Stereo | -0.75 | -0.87 | -1 | +1.19 | +10.38 | +10.29 |
| CREStereo | -5.17 | -6.88 | -1.24 | +14.92 | +16.7 | +82.35 |
| Selective-IGEV | **-0.53** | **+0.03** | -0.77 | +5.45 | +1.78 | +6.08 |
| BSV Ship | +0.68 | +74.65 | -2.9 | -4.36 | NKD | +5.36 |
| **OptiRSDE (Ours)** | -0.62 | -0.59 | 0.72 | **-0.04** | **+1.53** | **-1.77** |

Table 3: Depth estimation errors (in meters) for different methods. The values in the column headers represent ground truth distances (in meters). All listed methods are different approaches for estimating depth from calibrated left–right image pairs. **Bold** represents the smallest errors and underline represents the second smallest errors at a particular distance. **NKD** indicates cases where no keypoints were detected.

| Method | 8.20m | 16.70m | 24.19m | 44.26m | 47.16m |
|---|---|---|---|---|---|
| MonSter | +2.72 | +41.24 | +0.17 | +1.91 | **-0.17** |
| DSMNet | +0.54 | -0.45 | +0.37 | +1.03 | +0.47 |
| FoundationStereo | +0.46 | 0.5 | +0.44 | -0.75 | 0.61 |
| RAFT-Stereo | +0.81 | -0.26 | +0.51 | -0.29 | +1.07 |
| CREStereo | -5.62 | -10.93 | -15.32 | -21.57 | -22.32 |
| Selective-IGEV | +3.02 | +0.44 | +5.04 | +3.05 | +3.29 |
| BSV Ship | -0.16 | NKD | +1.23 | +47.26 | -1.4 |
| **OptiRSDE (Ours)** | **-0.01** | **-0.02** | **+0.08** | **+0.14** | +0.30 |

Table 4: Depth estimation errors on DrivingStereo's test images. Same structure as Table 3

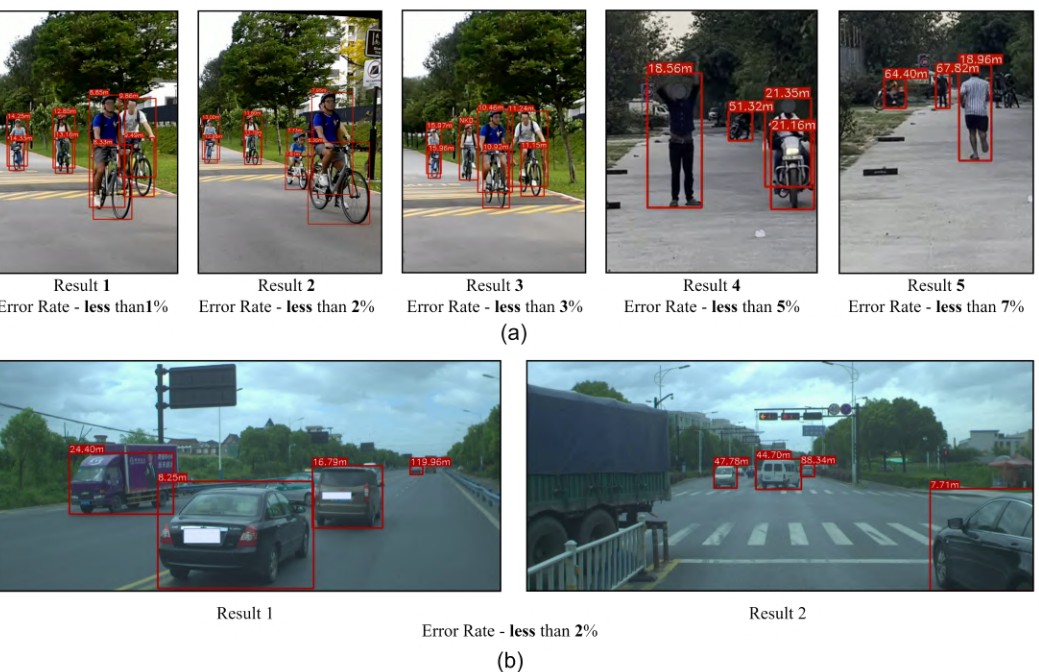

| | | | | |
|---|---|---|---|---|
| Result **1** | Result **2** | Result **3** | Result **4** | Result **5** |
| Error Rate - **less** than **1%** | Error Rate - **less** than **2%** | Error Rate - **less** than **3%** | Error Rate - **less** than **5%** | Error Rate - **less** than **7%** |

(a)

| | |
|---|---|
| Result 1 | Result 2 |

Error Rate - **less** than **2%**

(b)

Figure 4: Results of depth estimation up to 120m (a) from our own dataset. (b) from DrivingStereo's test images.

## 6 ABLATION STUDY

To evaluate the impact of individual components in our proposed pipeline, we conducted an ablation study by systematically disabling key modules and measuring their effect on depth estimation accuracy and computational efficiency.

- **Temporal Smoothing:** Removing the Exponential Moving Average (EMA) filter increased depth jitter, particularly at distances beyond 30m. Frame-to-frame variance rose from ±2% to ±10%, confirming EMA's critical role in stabilizing long-range estimates.
- **Feature Detection and Matching:** Replacing BRISK with ORB significantly decreased the robustness of keypoint detection. The results started showing more *No keypoint detected* errors, as ORB has difficulty detecting keypoints in relatively low-texture regions.
- **Synchronization:** Without QR-based synchronization, depth errors spiked up to 10% at 20m due to misaligned stereo pairs.
- **Computational Optimizations:** Disabling masking before keypoint detection, removing our keypoint matching optimization, or adding back FSRCNN significantly reduced the FPS (Table 5), while removing IQR-based outlier rejection increased depth variance by 30%.

| Method | FPS |
|---|---|
| Without KP detection masking | 4.85 (0.11) |
| Without KP matching optimization | 4.49 (0.08) |
| With FSRCNN | 1.53 (0.03) |
| **No ablation** | **5.38 (0.08)** |

Table 5: Execution performance after ablation. Mean FPS values with standard deviation in parentheses.

## 7 CONCLUSION

We presented OptiRSDE, a robust stereo depth estimation pipeline integrating temporal smoothing, optimized feature matching, and efficient synchronization to achieve high long-range accuracy and performance. Key contributions include its long-range precision, where pixel disparity refinement and harmonic averaging enable less than 10% depth error up to 100 meters; high performance, with several optimizations delivering about 5 FPS; and minimal setup, leveraging QR-based synchronization and chessboard calibration for simplified deployment. Ablation studies validated the necessity of each component, while benchmarks consistently demonstrated OptiRSDE's superiority over monocular and previous stereo baselines in terms of accuracy, speed, and robustness. The system's precise, long-range, fast, and temporally stable depth perception offers significant utility, poised to enhance autonomous driving (for accident detection and collision avoidance) and industrial safety systems. It can also support diverse industrial applications, including advanced robotics, object-aware automation, and sophisticated surveillance requiring accurate 3D scene understanding. Future work may involve exploring edge-device deployment and multi-sensor fusion (e.g., LiDAR) for robust performance in adverse weather conditions.

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

# 8 Appendix

This Appendix consists of an in-depth discussion of the following topics:

- Appendix A: QR Synchronization
- Appendix B: Keypoint detection masking optimization.
- Appendix C: Keypoint matching optimization.
- Appendix D: Visual comparison of results.

Due to the supplementary material size limit, images in this document will be of lower quality. The dataset samples and the results are also not provided in the zip file. Link for the full supplementary material is provided in the README.txt file.

# 9 Appendix A: QR Synchronization

Accurate stereo video frame synchronisation is crucial for reliable depth estimation. Since we did not have any cameras available with hardware synchronization properties, we devised a novel method to do automated software-based synchronization. A **QR code-based synchronization** method was implemented to precisely align left and right video streams.

At the start of each stereo video, sequential QR codes (e.g., 1, 2, 3, ...) are embedded into frames at the video's frame rate. During preprocessing, a QR code reader extracts these numerical values from each frame of both videos. Comparing these numbers reveals the frame offset (e.g., 14 or 20 frames) between streams. Based on this offset, unmatched or extra frames are discarded, resulting in perfectly aligned video streams. This significantly improves subsequent stereo processing accuracy by eliminating temporal misalignment and ensuring frame-to-frame correspondence, as shown in Figure 5.

Our method is designed specifically for stereo videos and works best when QR-based synchronization is used. For stereo videos without QR codes, the pipeline can still operate; however, synchronization errors typically around 10 to 12 frames can arise, which may significantly increase the depth estimation error rate.

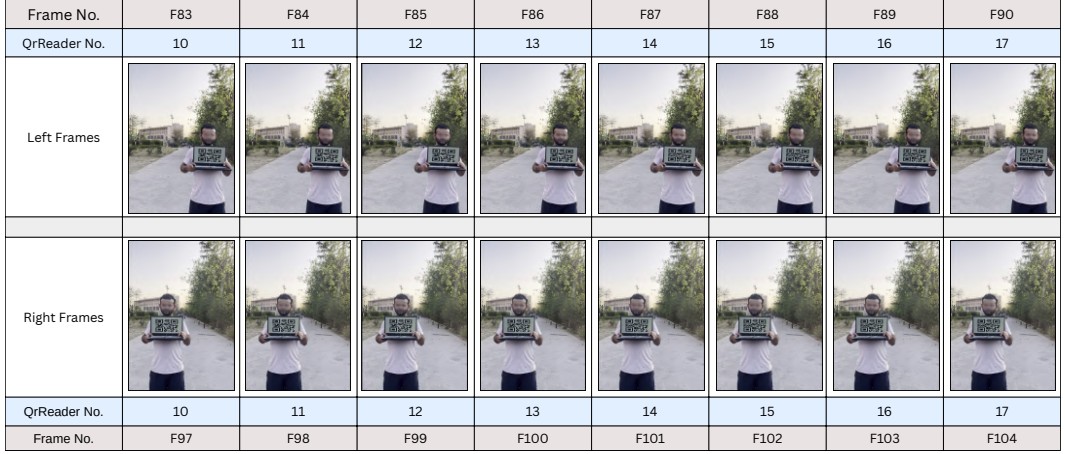

Figure 5: QR-based synchronization of stereo video frames. Sequential QR codes embedded in each frame allow precise alignment of left and right video streams by matching decoded numbers. This eliminates temporal offsets, ensuring accurate frame correspondence for improved depth estimation.

# 10 Appendix B: Keypoint Detection Masking Optimization

To optimize the computational efficiency of the BRISK keypoint detection algorithm within our OptiRSDE pipeline, we implemented a masking strategy. This targeted approach focuses keypoint

detection on relevant areas of the image, which significantly reduces processing time without compromising the integrity of the feature matching process.

## 10.1 OVERVIEW OF KEYPOINT DETECTION

Keypoint detection algorithms, such as BRISK, typically operate by exhaustively scanning the entire image to identify distinctive features. These features are robust to various image transformations and are crucial for tasks like feature matching. In a standard, unmasked approach, the algorithm processes every pixel, leading to considerable computational overhead, especially for high-resolution video. This results in a distribution of keypoints across the entire scene, including background elements that may not be relevant for object-level analysis. An example of such unmasked keypoint detection is presented in Figure 6.

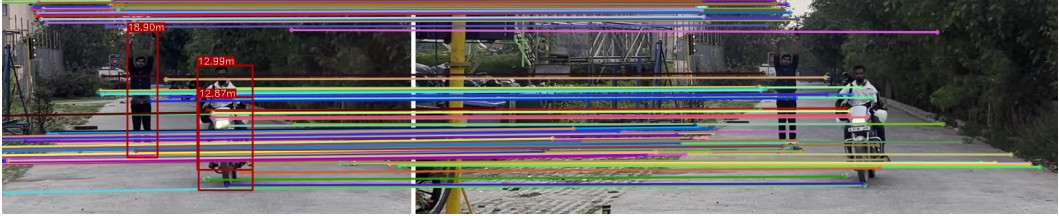

Figure 6: Illustration of unmasked keypoint detection and matching in left and right frames (separated by the white border). The left frame shows the detected objects and the estimated depths. Note the distribution of keypoints across both relevant objects and irrelevant background areas.

## 10.2 MASKED KEYPOINT DETECTION FOR EFFICIENCY

Instead of processing the entire image, we leverage the results from an initial object detection step (using YOLOv11 in our main pipeline) to define specific Regions of Interest (ROIs). For the purpose of object-level depth estimation, keypoints are primarily needed on and around the detected objects, not in static, irrelevant background regions of the image.

## 10.3 MASK GENERATION

For the left stereo image, a binary mask is generated directly from the bounding boxes provided by the object detector. This mask effectively restricts the BRISK keypoint detection algorithm to these object-containing regions, thereby substantially reducing the search space.

For the corresponding right stereo image, the masking strategy is critically adapted to account for the expected horizontal parallax shift inherent in stereo vision. An object appearing at a certain horizontal position in the left image will appear shifted to the left in the right image. Therefore, in addition to the regions defined by the left image's bounding boxes, we strategically expand the masked region in the right image to include the entire left side of each corresponding bounding box. This expansion is a crucial step to ensure that corresponding keypoints, potentially shifted due to disparity, are still well within the detection zone.

The effect of this masked keypoint detection is visually represented in Figure 7. Note that the blacked-out regions in the figure are for illustrative purposes only, and the mask is passed to the BRISK algorithm itself rather than being applied to the image pixels.

## 10.4 PERFORMANCE IMPACT

By focusing keypoint detection solely on these masked regions, the number of pixels that BRISK must process is drastically reduced. This direct reduction in computational load translates into a significant speedup for the keypoint detection phase of the pipeline. As detailed in the main paper's ablation study, disabling this keypoint detection masking resulted in a noticeable decrease in the overall Frames Per Second (FPS), confirming its indispensable role in achieving the high-speed efficiency of OptiRSDE. This optimization exemplifies how domain-specific knowledge can be effectively leveraged for substantial performance gains in computer vision tasks.

Figure 7: Illustration of masked keypoint detection and matching in left and right frames (separated by the white border). The left frame shows the detected objects and the estimated depths. Keypoints are concentrated within the object bounding boxes in the left frame and their expanded parallax regions in the right frame, demonstrating reduced processing area.

## 11 APPENDIX C: KEYPOINT MATCHING OPTIMIZATION

A primary computational bottleneck in keypoint-based stereo vision is the exhaustive matching process. A brute-force approach compares every keypoint in the left image against every keypoint in the right, leading to a combinatorial explosion of pairwise comparisons. This is computationally prohibitive for real-time applications, especially in dynamic scenes where low latency is critical. Our work, OptiRSDE, addresses this challenge by implementing an efficient, object-centric matching strategy that significantly curtails the search space.

Instead of a global, exhaustive search, our method localizes the matching process to relevant regions of interest (ROIs) defined by object bounding boxes. This targeted methodology drastically reduces pairwise comparisons by focusing computational effort only where needed. The cumulative effect of this optimization yielded a remarkable 16-fold performance improvement for keypoint detection and matching. In our experiments with a 10-second, 1080p video, this optimization reduced processing time from 110 seconds to a mere 6.9 seconds. This significant acceleration transforms a demanding task into a tractable one, paving the way for efficient, near real-time depth estimation for dynamic objects.

### 11.1 IMPLEMENTATION DETAILS

The core of our optimization is a two-stage filtering process. First, keypoints are detected across both the left and right frames using the BRISK algorithm. However, only keypoints within the predefined bounding boxes for each object are retained. All keypoints outside these regions are discarded, immediately reducing the dataset for the more intensive matching algorithm.

A critical aspect of this implementation is the correct handling of stereo parallax, similar to keypoint detection masking. Due to baseline separation, an object's projection in the right image shifts horizontally to the left. To account for this, the ROI in the right image is expanded, but unlike the masking stage, it is done for each object individually. While vertical bounds remain, the horizontal search area spans from the right edge of the box to the left edge of the image frame. This ensures all potential corresponding keypoints are included for matching, regardless of depth and disparity.

Only after this filtering and region adjustment is the brute-force matching algorithm, configured to use Hamming distance, applied. The matching is not performed globally on the filtered keypoints. Instead, it is done on a per-object basis, comparing keypoints from a left ROI exclusively against those in the corresponding adjusted right ROI. This targeted, object-by-object matching is a cornerstone of our optimized pipeline, ensuring both high speed and accuracy.

## 12 APPENDIX D: VISUAL COMPARISON OF RESULTS

In Figure 6, unmasked keypoint detection and stereo matching are shown. This visualization of stereo pairs shows keypoints scattered across both object regions and irrelevant backgrounds, potentially introducing noise and ambiguity in depth estimation. In contrast, Figure 7 showcases the stark contrast of masked keypoint detection, with keypoints tightly concentrated within object bounding

boxes and their projected parallax regions in the right frame, demonstrating reduced processing areas and more focused feature matching.

Moreover, a set of distance visualization frames is shown as Result 1, Result 2, and Result 3 in Figure 8, Figure 9, and Figure 10 simultaneously, where we compare depth estimation errors between OptiRSDE and the BSV Ship's baseline. The results clearly show that OptiRSDE achieves more accurate depth predictions across varying distances, including at challenging longer ranges where BSV Ship shows NKD (No Keypoint Detection) to detect keypoints. These comparisons visually and quantitatively highlight the spatial efficiency and superior depth accuracy of OptiRSDE.

Additionally, Result 4 in Figure 11 shows depth estimation errors on DrivingStereo's test images, further reinforcing the generalizability and performance of OptiRSDE on unseen data. The results in both datasets consistently demonstrate OptiRSDE's advantages over existing baseline methods, particularly in terms of its robustness and precision across various and challenging distance ranges.

The visualized stereo frame pairs correspond to the Result section of the main paper, including the depth estimation errors and the comparison between OptiRSDE and the BSV baseline methods. These visualizations offer a direct, intuitive understanding of the performance differences between the methods, giving a detailed visual comparison of the results presented in the Results section of the paper.

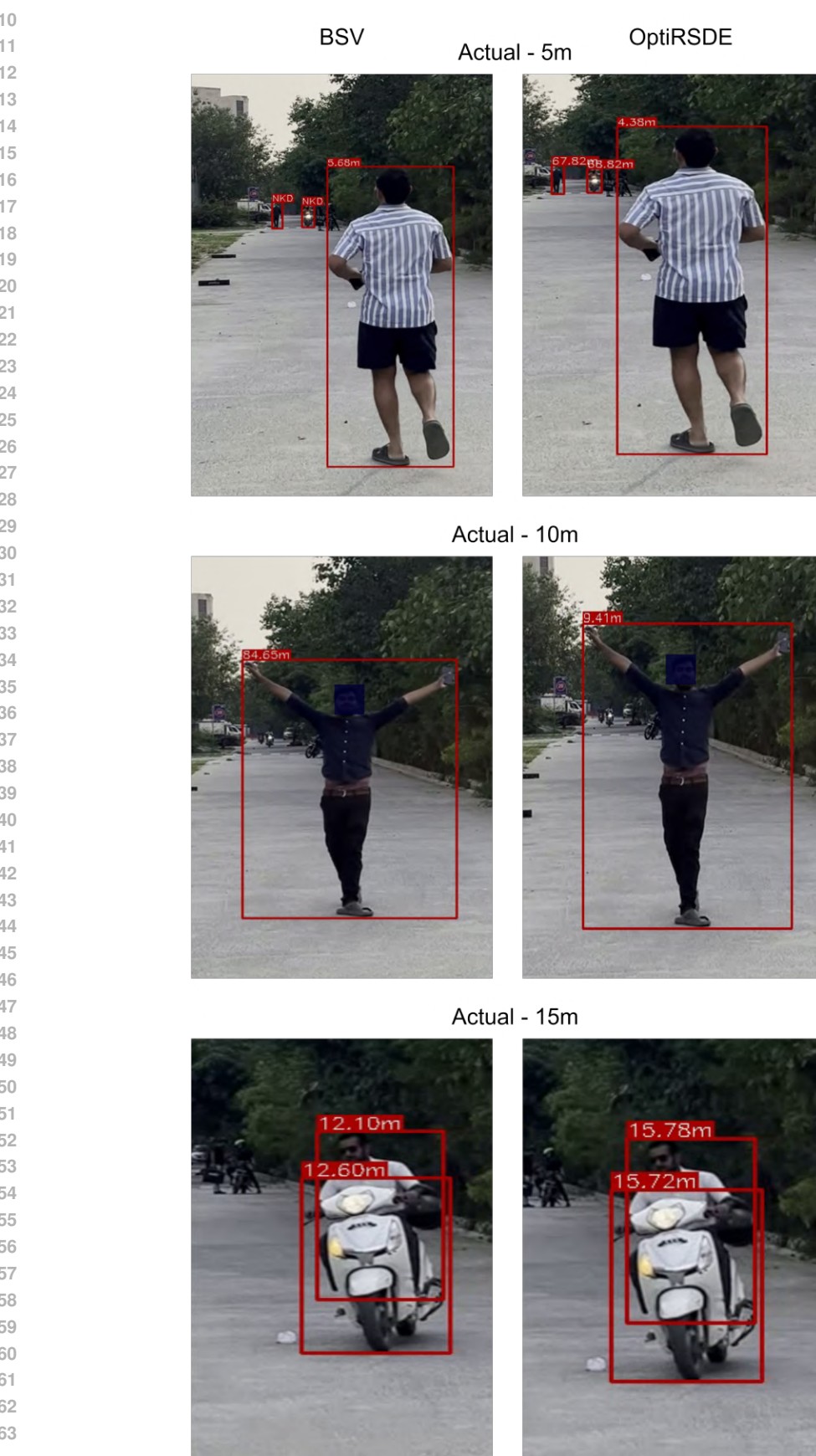

Figure 8: Result 1.

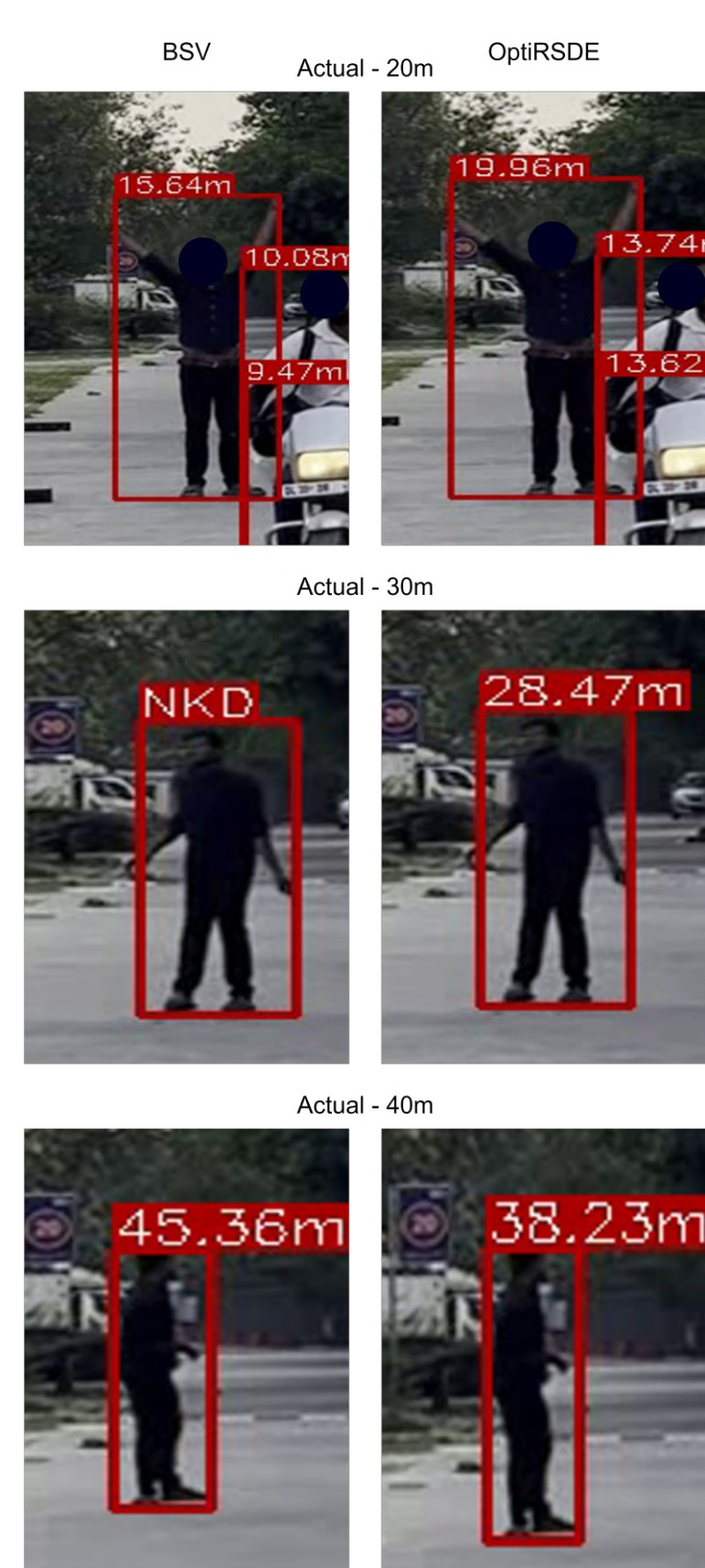

Figure 9: Result 2.

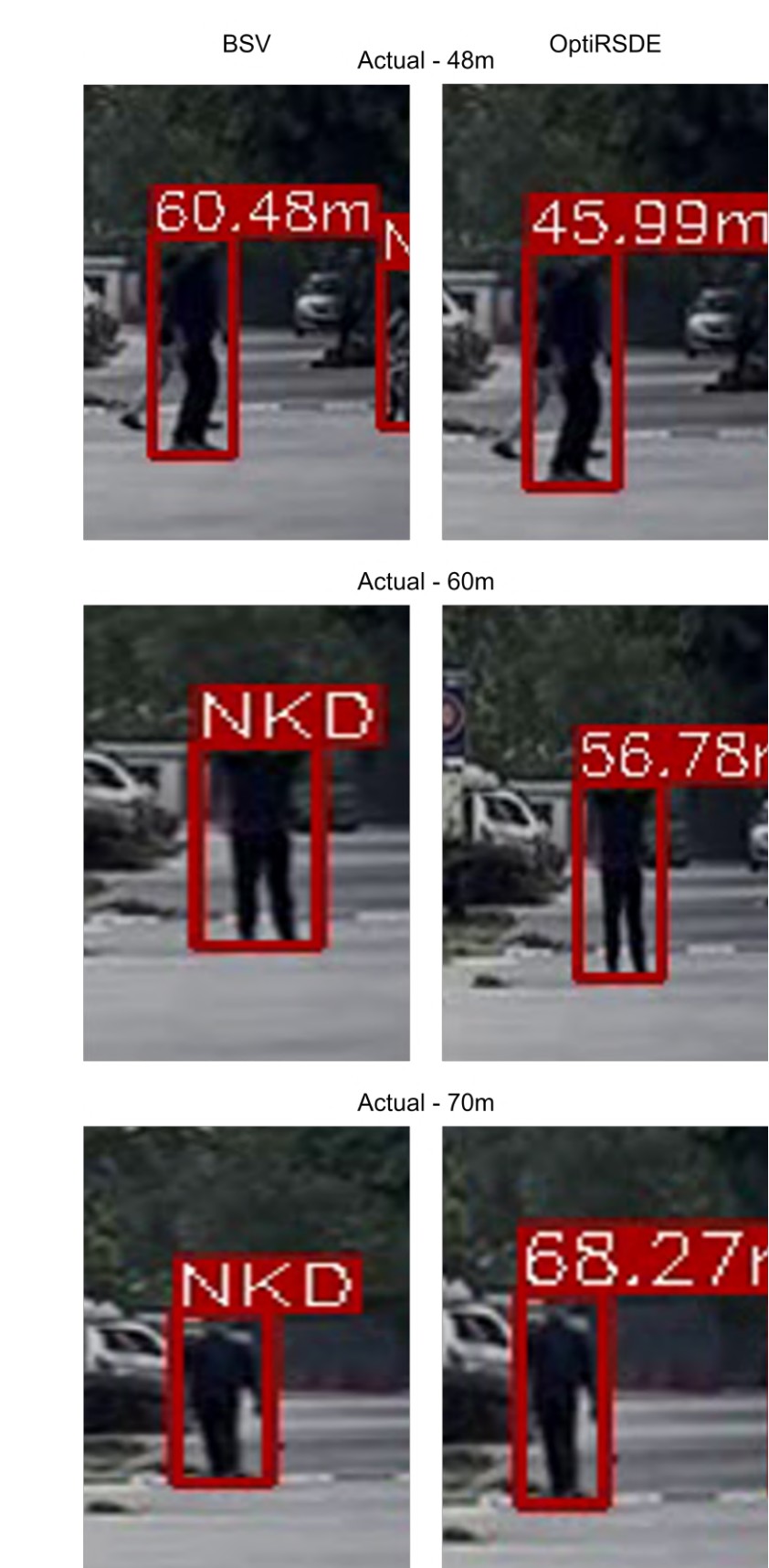

Figure 10: Result 3.

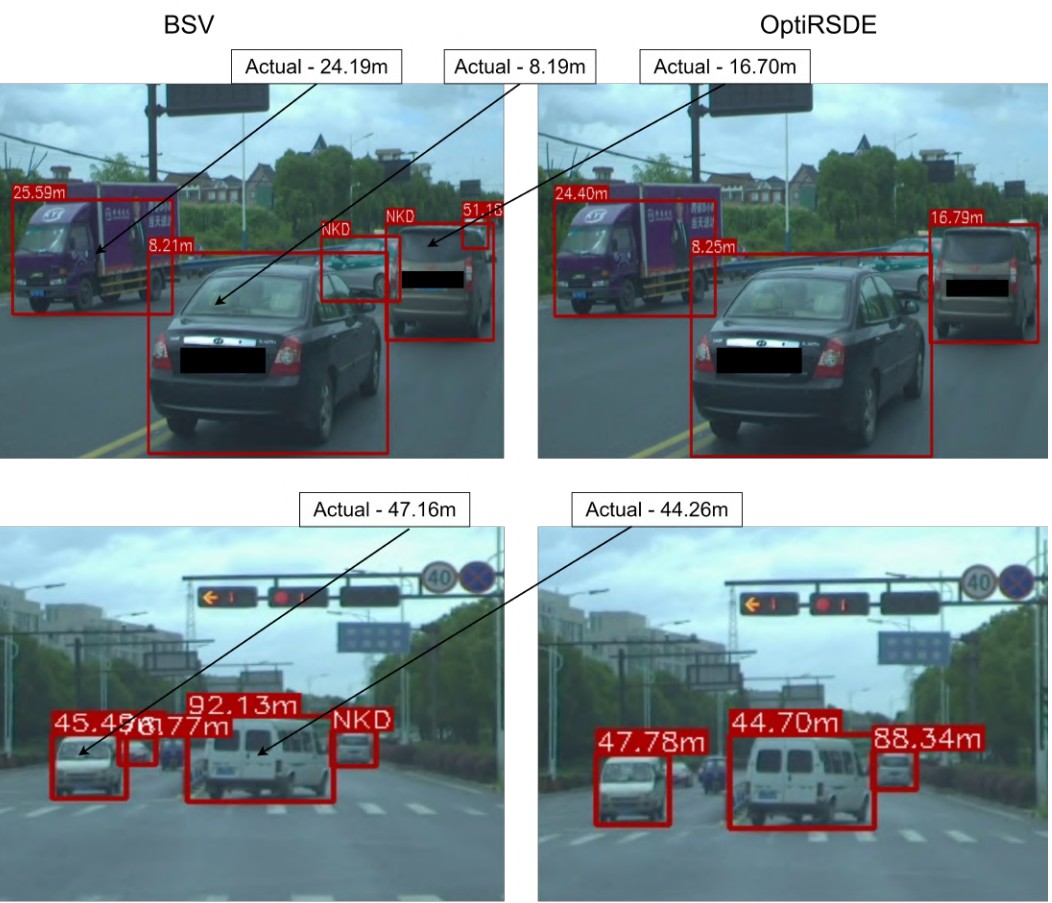

Figure 11: Result 4.

