# OpenReview forum: "OptiRSDE: A Novel Approach with Temporal Smoothing and Optimized Feature Matching for Fast and Robust Depth Estimation"
_ICLR.cc/2026/Conference — ICLR 2026 Conference Withdrawn Submission_

### Official Review · Reviewer_dmMJ · 2025-10-31

**Soundness:** 2
**Presentation:** 2
**Contribution:** 1
**Rating:** 2
**Confidence:** 5

**Summary:**

The authors propose OptiRSDE a classical approach for sparse depth estimation. The authors propose a full fledged pipeline starting from stereo camera synchronization, calibration, object detection, key point matching and depth estimatio through triangulation. The authors claim that their method outperforms MiDaS and multiple learning based stereo pipelines on a custom dataset and DrivingStereo.

**Strengths:**

A refreshing paper returning back to the classical geometry based depth estimaiton. The proposed method is an overview of a full fledged pipeline for depth estimation. The paper is well organized and well written.

**Weaknesses:**

Major weaknesses

1. There is nothing conceptually new in the proposed framework. All proposed steps are well known in the reserach community. The paper feels more like an article about a classical depth estimation approach with well engineered components than a research paper.

2. The comparisons in the paper are not fair/clear. The proposed approach is a sparse approach but it is compared against dense depth estimation pipelines. In table 3 and table 4 the depth buckets the authors pick to compare the methods are not consistent. Why the errors are signed? It is unclear if the numbers are given for single keypoints or they are buckets of keypoints that fall within some range.

3. The evaluation of the method relies on a small custom dataset and DrivingStereo. Using more enstablished datasets with benchmarks will strengthen the claims in the paper (Midleburry, KITTI, ETH3D)

4. The QR-code synchronization is more suited for a home based stereo system. There exist more rigorous methods to guarantee synchronization between cameras that can reach hardware level synchronization.

**Questions:**

Look at wekanesses

---

### Official Review · Reviewer_wHgJ · 2025-10-31

**Soundness:** 2
**Presentation:** 1
**Contribution:** 2
**Rating:** 2
**Confidence:** 5

**Summary:**

The paper introduces OptiRSDE, a lightweight binocular stereo depth estimation pipeline designed to achieve stable long-range performance in real-world environments. It addresses the limitations of existing methods in handling long-range depth estimation and high computational cost by introducing an object-level depth estimation design that improves both efficiency and stability. The system further integrates disparity refinement, temporal smoothing, and QR-code–based synchronization to enhance robustness. Overall, the work provides a practical and reproducible solution with valuable engineering insights for real-time, extended-range stereo depth estimation.

**Strengths:**

1. The paper identifies a clear and practical research gap, i.e., the need for real-time, binocular, and stable long-range depth estimation in real-world conditions. It clearly articulates the underlying challenges and proposes a well-structured and comprehensive pipeline to address them.

2. The introduction of object-level depth estimation notably enhances accuracy and efficiency, while reducing computational cost.

3. The authors demonstrate a good understanding of practical bottlenecks throughout the pipeline and propose targeted, context-aware solutions. For instance, they adopt BRISK-based feature detection and matching to improve robustness in low-texture regions. They also employ masking strategies to achieve effective computational optimization.

**Weaknesses:**

**1. Limited methodological novelty.** The paper raises a practically meaningful problem but lacks clear innovation at the algorithmic or theoretical level. The proposed system appears more as a complete engineering pipeline that integrates existing techniques at different stages rather than introducing fundamentally new principles. While the idea of object-level depth estimation is interesting, it currently reads as an implementation detail rather than a conceptual contribution. The authors could improve clarity by visually emphasizing this component in Figure 1, i.e., clearly separating the highlight idea or method of the paper from standard pipeline elements, and by explaining what unique problems it addresses.

**2. Insufficient description of the self-collected dataset.** The paper does not provide enough detail about the newly collected dataset, such as data acquisition process, sensors used, calibration procedure, data scale, environmental diversity, or its potential contribution to the community. Without such information, reproducibility and dataset value are difficult to assess. It is recommended that the authors include a dedicated section to introduce and describe their self-collected dataset in detail.

**3. Weak experimental support for key claims.** (**a**) The paper claims that OptiRSDE achieves superior long-range performance (< 3 % error at 50 m and 5–10 % at 100 m). However, quantitative results beyond 50 m appear only on the self-collected dataset, while in DrivingStereo the long-range results are merely qualitative. The reported quantitative evaluation extends only to 60 m, which does not substantiate the claim of robustness up to 100 m. It would strengthen the paper to include evaluations on additional public datasets that cover longer ranges. (**b**) The claim that BRISK performs better than ORB in low-texture regions is not convincingly demonstrated. The figures do not highlight cases where this advantage is visible or quantitatively measured. The authors are encouraged to include more representative and convincing qualitative results that better demonstrate the advantages of their method.

**4. Poor presentation and formatting issues.** The writing and presentation could be more structured and informative. For instance, in Section 4.1 (Experimental Settings), each subsection contains only one short sentence, which reads more like an experiment log. Merging these into a coherent paragraph would improve readability. Figures could also be more explanatory, e.g., in Figure 4, clarify why these particular examples are shown, what distinguishes them, and how they illustrate the method’s advantage. Consider adding RGB images alongside depth maps (full scene or focused object), color bars, and comparative results from other methods to better convey both the effectiveness and relative improvement of the proposed approach.

**Questions:**

1. Are the chosen evaluation metrics and experiments sufficient to support the paper’s claims? The authors state that the object-level design improves efficiency, yet no efficiency comparison is provided against other non–object-level depth estimation methods. Could the authors include such comparisons or clarify the actual computational advantages?

2. In Table 4, how were the specific distance values in the column headers determined? Please explain the rationale for selecting these exact numbers. Would it be clearer and more standardized to present the results using distance ranges (e.g., 0–10 m, 10–20 m, …) instead of discrete distance points?

---

### Official Review · Reviewer_EMC9 · 2025-11-01

**Soundness:** 1
**Presentation:** 1
**Contribution:** 1
**Rating:** 2
**Confidence:** 4

**Summary:**

This paper introduces OptiRSDE, a traditional vision-based stereo depth estimation framework optimized for speed, stability, and robustness in long-range scenarios (up to 100 meters). Unlike deep-learning-based approaches (e.g., RAFT-Stereo, FoundationStereo), OptiRSDE avoids neural networks and focuses on feature-based stereo geometry with efficient engineering refinements.
The method employs BRISK feature matching, QR-code synchronization, IQR-based outlier rejection, and exponential moving average (EMA) temporal smoothing. Additionally, YOLO-based object detection restricts computation to relevant regions, achieving real-time performance (5.38 FPS) while maintaining high accuracy (<3% error at 50m).
Experiments on DrivingStereo and custom datasets demonstrate strong robustness in low-texture and outdoor conditions, outperforming both monocular (MiDaS) and classical stereo methods.

**Strengths:**

1. Highly practical and deployable — The system can run in real time on standard hardware without GPU or model training, making it suitable for embedded or industrial systems.

2. Strong long-range accuracy — The method achieves under 3% error at 50m and 7–8% at 100m, surpassing both monocular and stereo learning-based baselines.

3. Temporal stability — EMA smoothing effectively suppresses frame-to-frame flicker and provides temporally consistent depth.

4. Innovative synchronization mechanism — QR-code-based synchronization provides a lightweight alternative to hardware sync, improving reproducibility.

5. Well-structured ablation analysis — The paper clearly quantifies the contribution of each module, demonstrating strong engineering rigor.

**Weaknesses:**

1. Limited research novelty — The contributions are mainly engineering improvements over conventional stereo pipelines, with little theoretical or algorithmic innovation.

2. Restricted to controlled environments — QR-code synchronization and manual calibration are difficult to apply in large-scale or outdoor autonomous systems.

3. No learning or adaptation mechanism — The system cannot benefit from data-driven improvement or cross-domain adaptation.

4. Limited support for dynamic scenes — The framework assumes static geometry and does not explicitly model object motion or parallax drift.

5. Questionable relevance in the era of foundation 3D models — With the rise of end-to-end reconstruction models like Dust3R, VGGT, and MASt3R, the necessity of continuing research on classical stereo pipelines becomes debatable.

6. Lack of visualization and failure analysis — The paper provides limited qualitative examples or failure cases to interpret model robustness.

**Questions:**

1. Scalability: Can the proposed system maintain its efficiency and accuracy when processing higher-resolution inputs (e.g., 4K) or multi-camera configurations?

2. Dynamic scenes: How does OptiRSDE handle moving objects or ego-motion drift without explicit motion compensation?

3. Synchronization practicality: Is QR-code synchronization feasible in outdoor or vehicle-mounted scenarios with varying lighting and occlusion?

4. Comparison with foundation models: In the era of end-to-end 3D foundation models (Dust3R, VGGT, MASt3R), why is a classical MVS pipeline still necessary? What unique advantages does OptiRSDE offer?

5. Reliability and safety: As the method is proposed for industrial and autonomous use, has any robustness or safety analysis been conducted under sensor failure or detection errors?

---

### Official Review · Reviewer_pusF · 2025-11-03

**Soundness:** 1
**Presentation:** 1
**Contribution:** 1
**Rating:** 2
**Confidence:** 4

**Summary:**

Paper proposes the use of Brisk keypoints constrained to detected object boxes for stereo depth estimation.
This is combined with temporal smoothing and a dynamic QR-code -based form of video synchronization.

They report substantially better numbers at 50 and 100m than other methods.

Key concerns:

1. The errors reported are *signed*. I don't know what this means, but you can't take the average of signed errors.
2. Errors reported depend on the keypoints, but different methods have different amounts and located keypoints and it is not a like-with-like comparison.
3. Unclear how video synchronization is handled for other methods. Are you using the same synchronization for all?

**Strengths:**

This is a sensible collection of heuristics that appear to work well. But I'm concerned that it is not a good fit for the conference (no learning is used in any components outside of a pretrained object detector), and I'm not sure how to interpret the results or if they make sense.

**Weaknesses:**

There's a lack of discussion about what the results measure, and several red flags.

1. The errors reported are *signed*. I don't know what this means, but you can't take the average of signed errors.
2. Errors reported probably depend on the keypoints (this is not described, and I'm guessing), but different methods have different amounts and located keypoints and it is not a like-with-like comparison.
3. Unclear how video synchronization is handled for other methods. Are you using the same synchronization for all?

The method will only estimate depth for keypoints in an object region identified by yolo.

**Questions:**

Can you explain what the result tables show, and respond to points 1 -3 in weaknesses?

As a presentational suggestion, I would move algorithm 1 to an appendix and increase the amount of discussion of the results.

---

### Note · Authors · 2025-11-12

**Comment:**

Not good comments

**Withdrawal Confirmation:**

I have read and agree with the venue's withdrawal policy on behalf of myself and my co-authors.